# Marine heatwaves disrupt ecosystem structure and function via altered food webs and energy flux

Dylan G. E. Gomes ⬤ [1,2,6] ✉, James J. Ruzicka ⬤ [3], Lisa G. Crozier ⬤ [4], David D. Huff ⬤ [5], Richard D. Brodeur ⬤ [5] & Joshua D. Stewart ⬤ [1]

The prevalence and intensity of marine heatwaves is increasing globally, disrupting local environmental conditions. The individual and population-level impacts of prolonged heatwaves on marine species have recently been demonstrated, yet whole-ecosystem consequences remain unexplored. We leveraged time series abundance data of 361 taxa, grouped into 86 functional groups, from six long-term surveys, diet information from a new diet database, and previous modeling efforts, to build two food web networks using an extension of the popular Ecopath ecosystem modeling framework, Ecotran. We compare ecosystem models parameterized before and after the onset of recent marine heatwaves to evaluate the cascading effects on ecosystem structure and function in the Northeast Pacific Ocean. While the ecosystem-level contribution (prey) and demand (predators) of most functional groups changed following the heatwaves, gelatinous taxa experienced the largest transformations, underscored by the arrival of northward-expanding pyrosomes. We show altered trophic relationships and energy flux have potentially profound consequences for ecosystem structure and function, and raise concerns for populations of threatened and harvested species.

Marine heatwaves (MHWs) are periods of prolonged, unusually warm ocean temperatures that can have significant impacts on marine ecosystems[1–4]. In tropical systems, sustained periods of warm water can cause coral bleaching and mass mortality events, which likely affects entire communities that rely on the complex structure and ecosystem functions provided by live coral[5]. In temperate systems, ocean temperature increases can lead to harmful algal blooms that produce toxins that kill other marine organisms[6,7]. These algae blooms can also lead to widespread hypoxic events, contributing to recent increases in the occurrence of ecological 'dead zones' that affect a wide range of species[8]. MHWs can alter nutrient cycling and availability in the ocean, which can affect the growth of phytoplankton. These bottom-up processes can alter lower trophic level productivity, which may in turn lead to stress and starvation in top predators, ultimately affecting reproductive success[9,10]. Cumulative impacts can lead to poleward distribution shifts of many pelagic species[11–13], resulting in disrupted or novel communities and changes in predator-prey relationships, which likely lead to changes in the overall structure of marine ecosystems as a consequence of MHWs. However, the full ecosystem-scale effects of MHWs have not been estimated within an

[1]Ocean Ecology Lab, Marine Mammal Institute, Department of Fisheries, Wildlife & Conservation Sciences, Oregon State University, Newport, OR 97365, USA. [2]National Academy of Sciences NRC Postdoctoral Research Associateship, Northwest Fisheries Science Center, National Marine Fisheries Service, National Oceanic and Atmospheric Administration, Seattle, WA 98112, USA. [3]Ecosystem Sciences Division, Pacific Islands Fisheries Science Center, National Marine Fisheries Service, National Oceanic and Atmospheric Administration, Honolulu, HI 96822, USA. [4]Fish Ecology Division, Northwest Fisheries Science Center, National Marine Fisheries Service, National Oceanic and Atmospheric Administration, Seattle, WA 98112, USA. [5]Fish Ecology Division, Northwest Fisheries Science Center, National Marine Fisheries Service, National Oceanic and Atmospheric Administration, Newport, OR 97365, USA. [6]Present address: Forest and Rangeland Ecosystem Science Center, United States Geological Survey, Seattle, WA 98195, USA. ✉e-mail: dylan.ge.gomes@gmail.com

impacted system to date, leaving substantial uncertainty in the short- and long-term consequences of MHWs on ecosystem structure and function.

The Northern California Current marine ecosystem extends from Vancouver Island, British Columbia, Canada to Cape Mendocino, California, United States. It is a highly productive upwelling marine ecosystem that supports high biomass of marine species, many of which are harvested in economically and socially important fisheries[14]. Sea surface temperatures in the ecosystem have been anomalously high in recent years, starting with the warm water blob in the winter of 2013-2014[15–17], which was among the largest and most intense MHWs ever recorded[1,18,19], and continuing through the present (Fig. 1a). The initial MHW starting in 2013 included temperatures roughly 3 °C above normal (exceeding three standard deviations) and persisted in two prolonged pulses until 2015[18,19]. Re-occurring MHWs, including the so called blob 2.0 in 2019, have since kept much of the North Pacific Ocean in a state of anomalously warm conditions over the past decade, indicating that these novel conditions are perhaps the new normal[18–20].

These anomalously warm periods have led to documented changes in the abundance and distribution of diverse taxa including gelatinous invertebrates, copepods, krill, squid, fishes and sharks[20–27], impacts to important fisheries[28], declines in primary productivity[29], increased abundance of relatively rare or non-existent Southern visitors (such as the sudden arrival of great abundances of a pelagic tunicate, *Pyrosoma atlanticum*)[24,30,31], and mass mortality events for seabirds and marine mammals[10,32]. Extreme warming events within the Northern California Current are expected to be exacerbated by climate change in complex and potentially non-linear ways[33,34]. Yet the ecosystem-wide ramifications of such sudden events are likely to be far greater than the expected changes due to long-term warming alone[35].

Here, we compare two end-to-end ecosystem food web models of the Northern California Current representing time periods immediately preceding (1999–2012) and following (2014–2022) the onset of a prolonged period of thermal anomalies (Fig. 1a) marked by at least two well-described marine heatwaves[36–38] to make inferences about ecosystem-level changes that have occurred since the onset of these recent extreme warming events. We estimate the effects of MHWs on the energy flow between producers and consumers across scales from individual functional groups to the entire food web network. Accounting for energy flux within the entire ecosystem allows us to directly estimate the cascading, ecosystem-wide effects of temperature-induced changes through both direct and indirect food web pathways.

## Results and discussion

Leveraging time series abundance data of 361 taxa (grouped into 86 functional groups, see Supplementary Data 1) from six long term surveys, diet information from a new diet database, and previous modeling efforts, we built two food web networks (pre- and post-onset of MHWs, hereafter pre-MHW and MHW) using an extension of the popular Ecopath ecosystem modeling framework (Ecotran[36,37,39]). Our comparative analysis of these two food web networks shows that lower trophic level biomass and energy pathways experienced greater changes after onset of MHWs than upper trophic levels (Figs. 1, 2), but that the energetic consumption (of lower trophic levels) and energetic contribution (to higher trophic levels) of many functional groups significantly changed between pre-MHW and MHW time periods (Figs. 3, S1–S5). Predators consumed prey both in different absolute quantities and proportions before compared with after the MHWs (Fig. 2).

By directly estimating changes in both biomass and trophic interactions across the entire ecosystem, we find that the largest perturbation to the energy flux of the Northern California Current ecosystem since the onset of multiple marine heatwaves is driven by a dramatic increase in the abundance of pyrosomes, *Pyrosoma atlanticum* (Fig. 1, Supplementary Data 1). This gelatinous species was essentially absent from the Northern California Current prior to recent MHWs[24,30,31,40], and this rapid increase drove substantial changes throughout the food web at low and mid trophic levels as pyrosomes consumed energy that would have been available for other groups (Fig. 1). Species at the base of the food web, such as pteropods, pelagic amphipods, small invertebrate larvae, small mesh-feeding jellies, krill, and sardine all consumed less phytoplankton in the MHW period, which may have contributed to decreases in their abundances. This, in turn, possibly left less forage for the carnivorous and larger jellies, which also declined (Figs. 1, 3).

Further, our models suggest that the majority of this re-directed energy does not flow to higher trophic levels, with >98% of pyrosome biomass ending up in detritus pools (Figs. 1, 2). Although there is evidence that some predators have consumed pyrosomes and other abundant gelatinous taxa during the MHWs[41,42], it is not clear what energetic benefits accrue to these predators compared to feeding on crustacean or fish prey. It has long been assumed that gelatinous prey are trophic dead ends, due to their low energy content[43], although recent advances in methodology suggest that gelatinous prey might be more important than previously believed[44]. In the Northern California Current, it appears that pyrosomes are not consumed as readily as jellies[36,41,42]. This may be because they are more difficult to digest, offer lower energy content, or remain novel to the food web such that predators have not yet responded. Thus, while overall gelatinous biomass in the ecosystem increased, the boom in scarcely consumed pyrosomes along with the concurrent decrease in jelly abundance has led to a marked decrease in the overall consumption of gelatinous prey since the onset of MHWs (Figs. 1, 2). More generally, a persistent shift toward filter-feeding gelatinous zooplankton and away from omnivorous euphausiids could have major negative implications for higher trophic levels including commercially important fishes, and thus food security in many ecosystems[45]. This further highlights the importance of understanding the uncertainty in the trophic influence of pyrosomes in their recently expanded northward range shift, the outcome of which will have important consequences for the future of the Northern California Current ecosystem under intensifying global warming.

Despite large shifts in biomass and connections of lower trophic levels (Fig. 1), the average trophic level did not change across models (Table 1). A higher number of trophic levels across a food web may signal a less efficient, or more unstable, ecosystem[46], while a low number of trophic levels can indicate a more efficient system as energy is lost at each level of consumption between the base of the food web and a higher trophic level species[47,48]. This may suggest that the efficiency of the Northern California Current food web is relatively robust, in the face of disruption by repeated MHWs. Various network metrics, such as connectance (the number of realized trophic links relative to the total possible number) and link density (the number of links per node) are measures of complexity that are also thought to relate to the robustness of food webs to disturbances[48–50]. Here, both network metrics were slightly higher in the MHW model, which might further suggest that the MHW ecosystem model is at least as robust to disturbance as the pre-MHW model was (Table 1). However, we show here that the pre-MHW ecosystem shifted considerably in response to repeated intense MHWs, despite also demonstrating metrics of network stability. As MHWs become more frequent and predictable on a global scale[51], it remains unclear how the Northern California Current ecosystem will be impacted by further perturbations or temporary reversals to pre-MHW conditions.

The current state of the ecosystem, characterized by increased pyrosome biomass and decreased energy flux to and from other low trophic-level species, may have important implications for fishery management. Chinook salmon and cod for example, appear to have

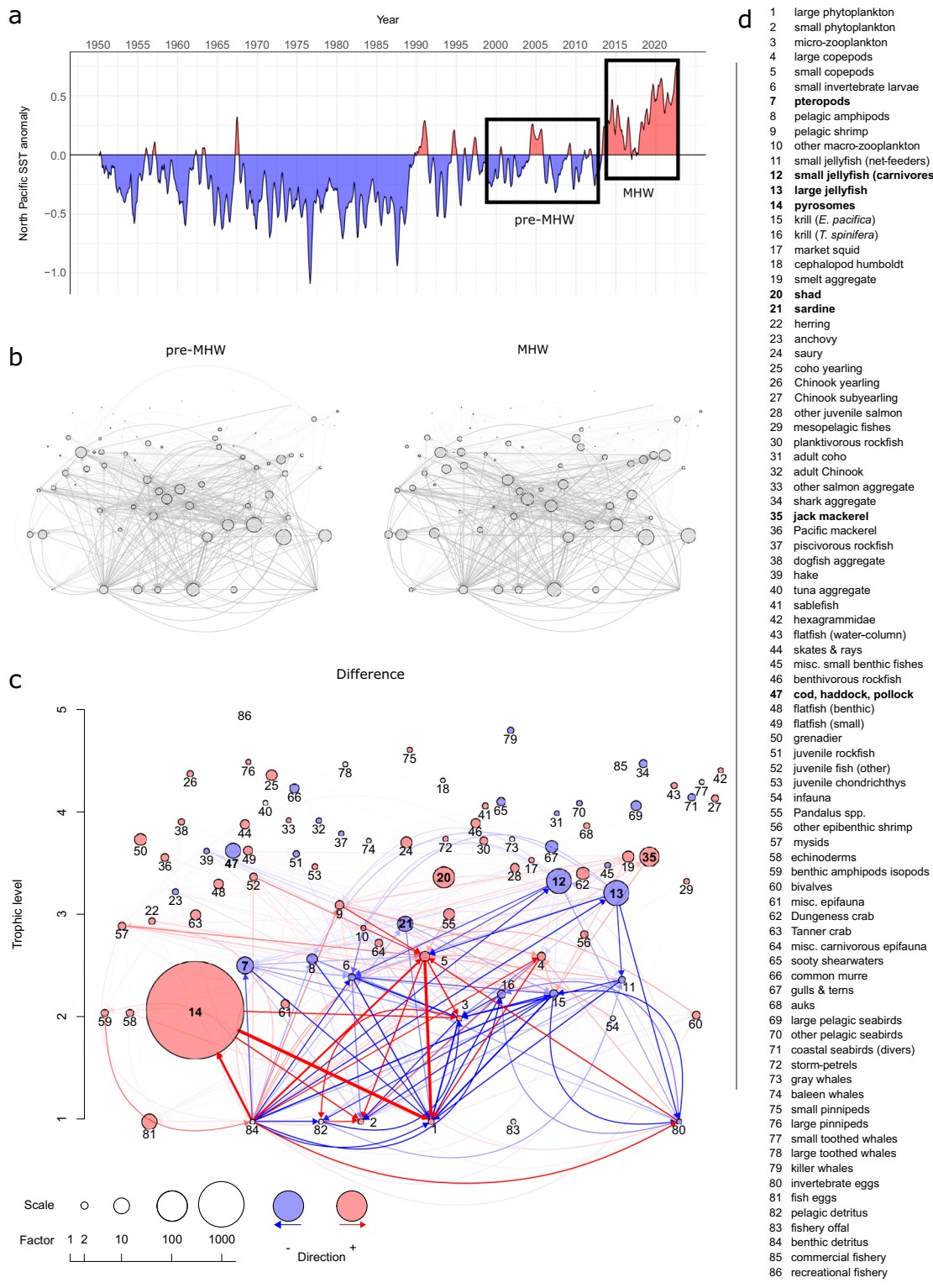

**d** A list of functional groups in ecosystem models:

1. large phytoplankton
2. small phytoplankton
3. micro-zooplankton
4. large copepods
5. small copepods
6. small invertebrate larvae
7. **pteropods**
8. pelagic amphipods
9. pelagic shrimp
10. other macro-zooplankton
11. small jellyfish (net-feeders)
12. **small jellyfish (carnivores)**
13. **large jellyfish**
14. **pyrosomes**
15. krill (*E. pacifica*)
16. krill (*T. spinifera*)
17. market squid
18. cephalopod humboldt
19. smelt aggregate
20. **shad**
21. **sardine**
22. herring
23. anchovy
24. saury
25. coho yearling
26. Chinook yearling
27. Chinook subyearling
28. other juvenile salmon
29. mesopelagic fishes
30. planktivorous rockfish
31. adult coho
32. adult Chinook
33. other salmon aggregate
34. shark aggregate
35. **jack mackerel**
36. Pacific mackerel
37. piscivorous rockfish
38. dogfish aggregate
39. hake
40. tuna aggregate
41. sablefish
42. hexagrammidae
43. flatfish (water-column)
44. skates & rays
45. misc. small benthic fishes
46. benthivorous rockfish
47. **cod, haddock, pollock**
48. flatfish (benthic)
49. flatfish (small)
50. grenadier
51. juvenile rockfish
52. juvenile fish (other)
53. juvenile chondrichthys
54. infauna
55. Pandalus spp.
56. other epibenthic shrimp
57. mysids
58. echinoderms
59. benthic amphipods isopods
60. bivalves
61. misc. epifauna
62. Dungeness crab
63. Tanner crab
64. misc. carnivorous epifauna
65. sooty shearwaters
66. common murre
67. gulls & terns
68. auks
69. large pelagic seabirds
70. other pelagic seabirds
71. coastal seabirds (divers)
72. storm-petrels
73. gray whales
74. baleen whales
75. small pinnipeds
76. large pinnipeds
77. small toothed whales
78. large toothed whales
79. killer whales
80. invertebrate eggs
81. fish eggs
82. pelagic detritus
83. fishery offal
84. benthic detritus
85. commercial fishery
86. recreational fishery

**Fig. 1 | Consumption matrix difference between pre-MHW and MHW food webs.**
**a** Sea surface temperature anomalies (°C) in the North Pacific from 1950 to 2022. Inset boxes indicate time period that pre-MHW and MHW ecosystem models are focused on. **b** Pre-MHW and MHW network diagrams show the food web consumption matrix. Trophic linkages (network edges) show rates of biomass exchange between trophic levels while the size of circles (network nodes) represent the absolute biomass densities in the system (on the log scale; see Supplementary Data 1). **c** A difference network was calculated as the difference between the pre-MHW model and the MHW model for both the edge weights and node biomasses. Node and edge sizes and colors depend on the magnitude and direction of change, respectively. Red colors indicate an increase from the pre-

MHW food web to the MHW food web, while blue colors indicate a decrease. The size of the circle corresponds to the magnitude of the change in biomass (see scale for multiplication factor, note that a factor of 1 means no change, and thus the circle will not appear) of a given functional group (indicated by the corresponding number, see Supplementary Data 1). Similarly, the thickness and color intensity of the lines (network edges) indicate the magnitude of change in energy flux between food webs. Node locations are identical in all three networks. The node numbers were omitted from the top two plots for easier visualization. **d** A list of functional groups in ecosystem models, with bolded names selected to highlight those with larger changes between model time periods. Source data are provided as a Source Data file.

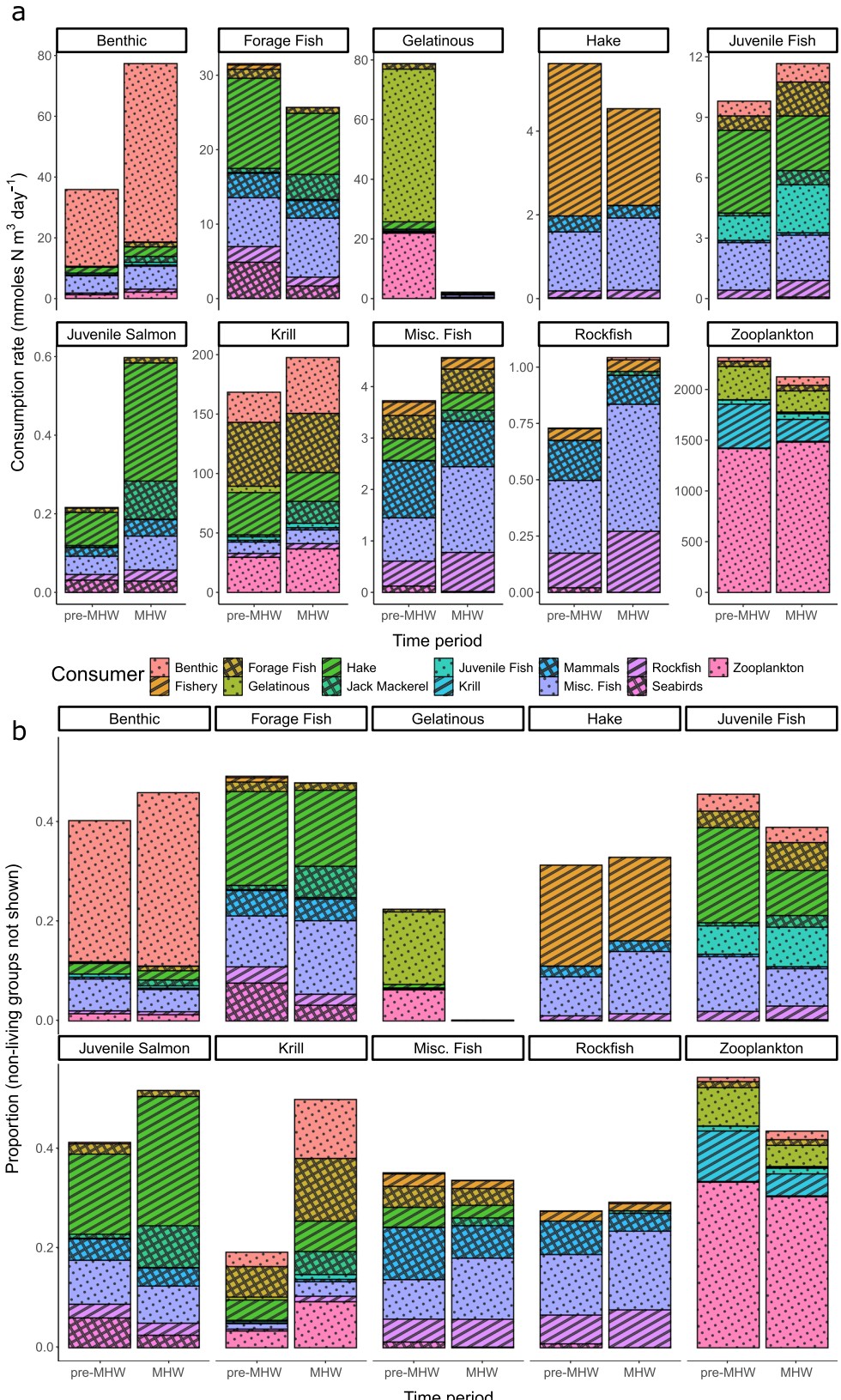

**Fig. 2 | Bottom-up energy flow through ecosystem functional groups.** The *x*-axis in both panels indicates the pre-MHW vs MHW time period. **a** The *y*-axis is the absolute consumption rate of specific functional groups (indicated in boxes across top of panel facets) by consumer groups (see legend for color-pattern combinations) in units of mmol N per cubic meter per day. **b** The y-axis shows the proportion of consumption that is allocated from specific functional groups to living consumer group types. All non-living nutrients and detritus pool groups were removed from these plots of energy transfer, because the ending fate of much of the system energy ends up in these pools, obscuring patterns in non-detritus groups. Thus, note that the *y*-axis in panel b does not extend to **1**, but the total proportion (including the non-living groups) still sums to **1**. A representative subset of taxa is presented across the figure facets. Source data are provided as a Source Data file.

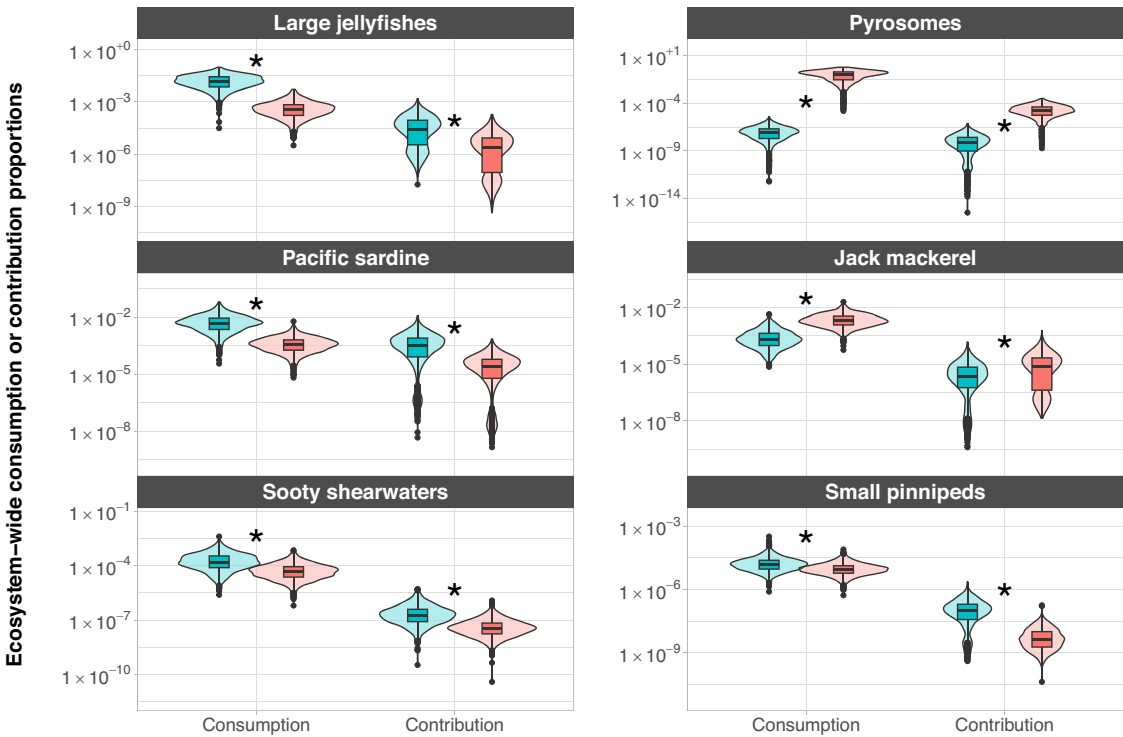

**Fig. 3 | Ecosystem-wide consumption and contribution values for representative taxa.** Violin plots show density of points from 1000 Monte Carlo runs of ecosystem models. Plotted over the violin plots, boxplots show median values as thick horizontal lines and first and third quartiles (the 25th and 75th percentiles) as the lower and upper edges of the box, respectively ($n = 1000$ independent Monte Carlo model parameterizations; see Methods). The lower and upper whiskers extend from the edges of the box to the values that are smallest and largest (respectively), yet no further than $1.5 \times$ interquartile range (i.e., the distance between the first and third quartiles) from the box. Outlying data beyond the end of the whiskers are plotted as individual points. Asterisks indicate that the difference in consumption of prey and contribution to predators between the pre-MHW and MHW models is significantly different (exact *p*-values found in Source Data file). Statistical significance was determined via *t*-tests with Bonferroni corrections for multiple comparisons. Units are proportions of total ecosystem consumption or contributions (see notes on footprint and reach in the Methods). See supplement for visualization of other functional groups. Source data are provided as a Source Data file.

decreased since the onset of the marine heatwaves in the Northern California Current (Fig. 1)[52]. Chinook salmon commercial harvest has been reduced by nearly a factor of three (corresponding to a 67% decline) in the Northern California Current since the onset of the MHWs [PacFIN, http://pacfin.psmfc.org][36,37]. Cod do not sustain a major fishery in the Northern California Current, but they are an important fished species in Alaska, where declines have similarly been documented during these recent North Pacific MHWs[4]. If these species do not return to their former abundance and biomass, commercial fisheries may have to shift their efforts towards more readily available species. The increased dominance of Pacific jack mackerel (*Trachurus symmetricus*) in the Northern California Current ecosystem is demonstrated by huge increases in their abundance in recent years[24,53] (Fig. 1). The impact of jack mackerel on lower trophic levels has increased since the onset of recent marine heatwaves (Figs. 2, 3). Yet, despite this increase in abundance, commercial fisheries in the U.S. have not shown any change in jack mackerel landings [PacFIN, http://pacfin.psmfc.org/][36], which might suggest jack mackerel is an under-utilized resource that can support substantial fishery landings. Adapting harvest strategies to account for changes in ecosystem structure could represent a significant step towards climate-resilient fisheries. However, further work is needed to determine if changes in the abundance of these species are directly influenced by marine heatwaves. Pacific sardine (*Sardinops sagax*) have complicated population fluctuations in response to multiple factors, and are often thought to have a positive relationship with

ocean temperatures[54,55]. Yet sardine collapsed just prior to the onset of the MHWs, and populations have not shown signs of rebuilding despite the warm ocean conditions persisting throughout the California Current Ecosystem[56].

Ecosystem modeling tools remain under-utilized in exploring the impacts in large scale disturbances such as marine heatwaves. The work presented here is a static comparison across measured ecosystem model states, yet future dynamic modeling can include abiotic variables such as ocean temperatures and mechanistic links to physiological rates across members of the food web. A 2 °C change in water temperatures can lead to an estimated 3.7% increase in the rate of biomass flow to higher trophic levels (also known as the biomass turnover rate and the production to biomass ratio, P/B[57]). This means that climate change may be exacerbating the already global trend of faster biomass transfers due to fishing pressure[58]. Faster biomass flow can indicate lower overall biomass in an ecosystem as the biomass residence time within each trophic level is reduced[58]. Furthermore, using general $Q_{10}$ scaling relationships[59] a 2 °C increase in temperature can lead to an increase in metabolic rate of about 14.9% and a lower production efficiency (less energy is directed to production of new biomass as metabolic rates increase). This can result in a reduced trophic transfer efficiency, or the fraction of energy consumed by prey that is transferred to consumers at the next higher trophic level. The ecosystem then becomes less efficient at supporting higher trophic levels. Functional groups with

## Table 1 | Network metrics

| Statistic | Pre-MHW | SD | MHW | SD | *p*-value | *t*-value | df |
|---|---|---|---|---|---|---|---|
| Number of nodes | 86 | – | 86 | – | – | – | – |
| Mean Trophic Level (TL) | 3.268 | 0.995 | 3.288 | 1.030 | 0.90 | −0.128 | 169.8 |
| TL weighted by biomass | 2.339 | 0.995 | 2.413 | 1.030 | 0.63 | −0.481 | 169.8 |
| Connectance | 0.229 | 0.022 | 0.251 | 0.023 | 2.15 ×10−10 | 6.698 | 197.8 |
| Link density (links per node) | 19.709 | 1.875 | 21.605 | 1.942 | 2.15 ×10−10 | 6.698 | 197.8 |

Comparison between pre-MHW and MHW ecosystem model network values with associated standard deviations (SD). Two-tailed *T*-test outputs are reported as *p*-values, *t*-values, and degrees of freedom (df). N = 100 independent Monte Carlo model parameterizations (see Methods). Connectance = the number of (non-zero) realized links relative to the total number of possible links (86 × 86 = 7396). None of the calculations include nutrients as functional groups.

particularly low initial production efficiencies in the pre-MHW model (e.g., jack and Pacific mackerels) would need to increase their consumption rates to maintain viable populations in a warmer, more metabolically costly environment. It is unclear, however, whether the production of lower trophic level species could support a sustained increase in predation pressure in combination with existing fishing pressures.

As marine heatwaves intensify and are increasingly common, these climate disturbances will have major impacts on marine ecosystems[2] with both winners and losers as some species take advantage of changing conditions or expand their distribution whereas others struggle to adapt or are physiologically constrained[60]. Estimating changes to energy flow within ecosystems and the energetic consumption and contribution of individual functional groups provides an approach for quantifying ecosystem changes as global warming increasingly disrupts food webs and the sustainable use of such resources. Ecosystem models may provide a means for better predicting future winners and losers as we prepare for climate resiliency in ecosystem-based fisheries management and the recovery of threatened and protected species in ecologically and economically important ecosystems.

## Methods

### EcoTran

Pre-MHW and MHW were built and analyzed within the EcoTran end-to-end ecosystem model platform[39]. EcoTran builds upon the widely-used Ecopath food web modeling framework[61]. One NCC ecosystem model was parameterized from datasets collected prior to the 2014 onset of MHW[37] and the other model was developed recently, which is based on datasets from 2014 onwards through multiple warm ocean years[36]. Both models represent 80 living functional groups, 3 nutrient pools, 5 detritus pools, and 2 fisheries, which are parameterized with multiple sea surveys (biomass)[22,53,62–66], commercial and recreational fishery databases (landings)[67,68], a trophic interaction database (diets)[69], and various sources of literature, among other unpublished data (described in more detail in Gomes et al., 2022[36] and Ruzicka et al., 2012[37]). The trophic interactions within each model are described as a production matrix defining the fate of all consumption by each group between its metabolic costs, non-assimilated egestion, biomass production that is consumed by each predator or fleet, and senescence[36,37,70]. For the analyses presented here we use static, steady-state ecosystem models, which has the advantage of allowing for a direct comparison across ecosystem time periods (parameterized with data collected within this system) without requiring a complete understanding of how MHW physical variables (e.g., temperatures) mechanistically affect each functional group's physiology and trophic relationships.

### Sea surface temperature (SST) anomalies

Monthly SST anomaly values for the North Pacific Ocean (Fig. 1a) are from the Met Office Hadley Centre HadSST data set[71,72]. Data were plotted in R[73] with help from 'ggplot2'[74].

### Model adjustment details

The pre-MHW model, adapted from Ruzicka et al. (2012)[37], was updated in several ways. First, Ruzicka et al. (2012) used Ecopath methods to estimate the biomass of euphausiids required to to maintain the pre-MHW food web in thermodynamic balance[37]. Since then, a longer and more precise euphausiid time series has been developed from the Joint U.S.-Canada Integrated Ecosystem and Pacific Hake Acoustic Trawl Survey and modeling efforts[63]. In the MHW model, Gomes et al. (2022)[36] used euphausiid biomass densities from the 2015, 2017, and 2019 surveys[63]. The pre-MHW model was updated to use the earlier period of the time series (2007, 2009, 2011, and 2012). Similarly, Ecopath-based estimates of invertebrate egg biomasses were dramatically different between the pre-MHW and MHW models[36,37]. We used survey data from the Newport Hydrographic Line (NHL)[65] to parameterize invertebrate eggs in the pre-MHW (NHL data: 2000–2012) and MHW (NHL data: 2014–2020) models. Due to the changes in biomass between the previously published pre-MHW model and the one used here we needed to re-balance the ecopath model because phytoplankton were slightly over-consumed by the increase in euphausiids. We accomplished this with small changes to the diet matrix of the pre-MHW model to move pressure off of phytoplankton (see supplemental preMHW_DietChanges.csv). See previous food web models for more information about data used, model-building, and mass-balancing procedures[36,37].

Due to slight differences in the food web model structure between the pre-MHW and MHW ecosystem models, it was necessary to combine some functional groups to make comparisons across the models (see Supplementary Data 1). For all groups that were combined, diets were aggregated as a weighted averaging (weighted by biomass of individual components/species of the functional group). Total functional group biomasses (for mass-balancing) were summed across the constituent components/species within each functional group. Pyrosomes are not thought to have been present in the pre-MHW period in the NCC ecosystem[31], but to make models directly comparable, we added a trace amount of pyrosomes to this pre-MHW model (0.00001 mt/km2). Marine mammals from Gomes et al. (2022) were combined to match that of Ruzicka et al. (2012); that is, Northern elephant seals and sea lion (California and Steller's) functional groups were combined into a large pinniped group and other killer whales and southern resident killer whales were grouped into a killer whale functional group (as they both originally were in Ruzicka et al., 2012). Juvenile salmon groups also did not match between the two ecosystem models; for simplicity, they were combined into four groups within each model: yearling coho, yearling chinook, subyearling chinook, and other juvenile salmonids. Similarly, all commercial fleets were aggregated into one commercial fishery fleet due to recent changes in the PacFIN database fleet names (http://pacfin. psmfc.org/)[67]. To see expanded fleet information, please see Ruzicka et al. (2012) and Gomes et al. (2022).

### Network analyses and metrics

To compare food web structures pre-MHW and MHW, we measured average (arithmetic mean) trophic level, average trophic

level weighted by the biomass of each functional group, network connectance (the number of non-zero realized links relative to, i.e., divided by, the total number of possible links; 7396 in an 86 × 86 network), and link density (the average number of connecting links per functional group) (Table 1). The initial food web for the MHW model was built upon the pre-MHW model, such that pre-MHW trophic connections were included in the MHW along with new diet data[36]. To ensure that the network connectance and link densities were not artificially inflated in the MHW period, we removed all trophic connections within the MHW network that were originally carried over from the pre-MHW model (and which were not represented in the updated diet data; see supplemental code).

A difference network was calculated as the difference between the edges and node biomasses in the pre-MHW model and the MHW model (Fig. 1). Node and edge sizes and colors are dependent on the magnitude and direction of change, respectively. Blue signifies a decrease in (from pre-MHW to MHW) biomass density (nodes) or energy flow (edges) and red signifies an increase. Bigger circles indicate (on a log scale) higher differences in biomass and thicker and darker edges denote larger changes in energy flows. The network was visualized with help from the R package 'qgraph'[75].

### Energetic consumption and contributions

To make comparisons of ecosystem structure between pre-MHW and MHW years, steady-state models were used to estimate the relative importance of functional groups for transferring energy to higher trophic levels. We calculated the relevance of targeted functional groups as both consumer (consumption of lower trophic levels) and producer (contribution to next trophic level) with two non-dimensional metrics: footprint and reach, respectively.

A functional group's trophic impact upon lower trophic levels is expressed by its footprint, which is the fraction of each producer group's total production supporting focal consumer groups via all direct and indirect pathways (excluding detritus). The footprint, in other words, is the proportion of energy from lower trophic levels (relative to a focal group) consumed by that focal group. Conversely, the importance of a focal functional group to higher trophic levels was expressed by its reach: the fraction of consumers' production that originated with (or passed through) that functional group via all direct and indirect pathways[37]. Thus, the reach is the proportion of energy consumed by higher trophic levels (relative to a focal group) that passed through that functional group.

Footprint and reach can be defined broadly (i.e., the footprint upon all lower trophic levels) or precisely (i.e., the footprint upon one specific producer). For our general ecosystem-wide comparison of the roles of phytoplankton, copepods, euphausiids, forage fishes, gelatinous zooplankton, rockfishes, and fish, seabird, and mammalian predators, we adopted the broadest definitions, considering footprint and reach relative to (proportions of) total system production and total consumer production, respectively.

The net uncertainty among physiological parameters, diet, and nutrient cycling terms are expressed as levels of uncertainty about each element of the production matrix. In our analyses, each element of the production matrix was randomly varied by drawing model parameters from a normal distribution with a mean of the originally parameterized value and a standard deviation [converted from conservative coefficient of variation (CV) values, which were based on the Ecopath pedigree strategy of assigning uncertainty based on the types of (i.e., survey type) and confidence in (quantity and quality) our data sources; see supplemental data and code repository readme]. For example, we used diet information from 39083 individual juvenile Chinook salmon, 2911 Pacific herring, and seven pyrosome colonies[36]. Thus, we set CV values for each element of their diet vectors to 0.1, 0.5, and 0.8, respectively to reflect differences

in the robustness of the datasets (and the associated uncertainty). We drew 1000 possible Monte Carlo food web models to investigate the propagation of uncertainty for footprint and reach of each assessed functional group within each model (pre-MHW and MHW). Footprint and reach values were plotted with help from the R package 'ggplot2'[74].

### Estimates of biomass flow and trophic transfer efficiency changes due to temperature

The inverse of the residence time of biomass, or the speed of biomass flow, found in Maureaud et al. (2017)[58] is defined as the production to biomass ratio (P/B) for any particular group and allows for the inclusion of temperature (T) as:

$$\left(\frac{P}{B}\right) = 1.06 \times e^{0.018 \times T} \times K^{0.75} \tag{1}$$

Assuming no change in the von Bertalanffy asymptotic growth rate parameter (K), the ratio between MHW (P/B) and pre-MHW (P/B) simplifies to:

$$\frac{e^{0.018 \times T_{MHW}}}{e^{0.018 \times T_{preMHW}}} \tag{2}$$

Or more simply:

$$e^{0.018 \times T_{MHW} - 0.018 \times T_{preMHW}} \tag{3}$$

Using a hypothetical temperature change of 2 °C we calculated a scaling factor of 1.037 (increase in 3.7%) to convert P/B values parameterized during pre-MHW conditions to an ecosystem that is 2 °C warmer. To explore changes that might occur to trophic transfer efficiencies due to higher metabolic costs associated with higher temperature, we used classic $Q_{10}$ scaling relationships. We estimate the proportional change in metabolic rate (M) as:

$$\frac{M_{MHW}}{M_{preMHW}} = Q_{10}^{\Delta T/10} \tag{4}$$

where $\Delta T$ is the change in temperature between MHW and pre-MHW ecosystems and $Q_{10}$ is a temperature scaling coefficient[76]. We used a common and general $Q_{10}$ value of two[77] for calculations of the metabolic scaling factor of 1.149 (14.9%). As metabolic costs increase, trophic transfer efficiencies decrease.

### Statistical analysis

Two-tailed $t$-tests in R[73] were used to compare mean trophic levels, network connectance, link density, and the footprint and reach metrics (for each functional group) across pre-MHW and MHW models. Since we made multiple comparisons for footprint and reach metrics (each functional group × both footprint and reach metrics), we corrected $p$-values with conservative Bonferroni corrections. For network connectance and link density, we created 100 bootstrapped networks by randomly sampling, with replacement, which nodes to use, each iteration re-calculating connectance and density, which were then compared across models.

### Reporting summary

Further information on research design is available in the Nature Portfolio Reporting Summary linked to this article.

## Data availability

All data and materials used in the manuscript are available in a long-term data repository at: https://doi.org/10.5281/zenodo.8121889. Source data are provided with this paper.

## Code availability

All MATLAB (R2021a) and R (v 4.2.2) code used in the analysis are available in a long-term repository at: https://doi.org/10.5281/zenodo.8121889.

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

## Acknowledgements

We thank Mary Hunsicker and Andrew Thompson for feedback on earlier versions of this manuscript. We thank Elizabeth M. Phillips, Alicia Billings, Cheryl A. Morgan, Jen E. Zamon, Elizabeth A. Daly, Joseph. J. Bizzarro, Jennifer L. Fisher, and Toby Auth for access to biological survey data, and Isaac Schroeder for supplying SST anomaly data for Fig. 1a. The National Academy of Science's National Research Council Postdoctoral Research Associateship Program provided funding to DGEG.

## Author contributions

Conceptualization: D.G.E.G., J.J.R., L.G.C., D.D.H., R.D.B., J.D.S. Data curation: D.G.E.G., J.J.R. Formal analysis: D.G.E.G. Funding acquisition: L.G.C., D.D.H., D.G.E.G. Investigation: D.G.E.G. Methodology: D.G.E.G., J.J.R. Project administration: D.G.E.G., J.J.R., L.G.C., D.D.H. Software: D.G.E.G., J.J.R. Supervision: D.G.E.G., J.J.R., L.G.C., D.D.H., J.D.S. Validation: D.G.E.G. Visualization: D.G.E.G. Writing—original draft: D.G.E.G. Writing—review and editing: D.G.E.G., J.J.R., L.G.C., D.D.H., R.D.B., J.D.S.

## Competing interests

The authors declare no competing interests.
