## [Peer Review File · Nature Communications]

Marine heatwaves disrupt ecosystem structure and function
via altered food webs and energy fluxREVIEWER COMMENTS

Reviewer #1 (Remarks to the Author):

This manuscript uses modelling to explore the ecosystem-level impacts of persistent marine heatwaves in the Northeast Pacific Ocean. More specifically, the authors explore the differences between two Ecotran models fit to data from the region pre- and post- onset of the marine heatwaves to evaluate changes in the structure of the ecosystem in the Northeast Pacific Ocean in response to heatwaves. They find noticeable changes in energy flow and composition of the ecosystem pre- and post- heatwave. I thought the paper was easy to understand and interesting, exploring a topic of likely significant interest to a wide audience. I have one significant comment, and then some minor comments about presentation.

Significant comment:

From what I understand, the structure of the ecosystems as represented in the Ecotran models are static, since they are fit to data from the system they're representing (Northeast Pacific Ocean pre- and post- heatwave). This means that the primary linkage between the pre- and post- heatwave models are through the data; the post- heatwave model configuration does not emerge from the pre- heatwave model, in response to changes in environmental conditions. To me, this is a significant limitation that should be discussed in the main text, because I think it makes the arguments around novel regime shifts, and the stability of the post-MHW system unconvincing (lines 121-140). The pre-MHW system changed in response to shifting environmental conditions, to a post-MHW structure that has similar metrics of stability as the pre-MHW system (Table 1). But, what's stopping the post-MHW system (in the real world, not in the model) from moving to an alternative state again, in response to even greater warming in the future? If the pre-MHW changed, even though it wasn't unstable, why wouldn't the post-MHW system change again? This is a significant limitation of the modelling approach I think (unless I'm missing something); the model ecosystem structure is baked in, it can't change in response to environmental conditions, even though the real system changed and may (will?) change again. The post-MHW model does not emerge from the pre-MHW configuration, the only link between the two is that the observations come from the same system. This doesn't mean the results aren't

interesting, or the comparison isn't sound, but it means that I don't think you can talk about the stability of regime shifts convincingly, since an already stable system (by your metrics in Table 1) shifted in response to environmental conditions.

Minor comments:

Line 153 - the reference to Figure 1 to discuss the increased dominance of jack mackerel is almost impossible to find in the Figure. I suggest highlighting the group differently, or finding another way to reference the change.

Line 254-259 - footprint/reach sounds like the same thing to me, from what's written. An example of each would really help.

Line 269 - what is a "standard" values for the CV?

Figure 1 - what is the unit for "the magnitude of the change in biomass", i.e., the size of the circles? A reference circle is needed for the delta plot.

Figure 2b - y-axis label needs to be more informative. When I see "proportion" I ask why it doesn't go from 0 to 1. It's explained in the figure caption but it should be easy to understand from the figure itself.

Figure 3 - what are the units for "ecosystem-wide consumption and contribution values for representative taxa"? Are these percentages?

Reviewer #2 (Remarks to the Author):

This study aims at quantitatively describing changes in the structure of a marine ecosystem due to a marine heatwave (MHW). The topic is highly relevant since MHW are increasing in frequency, in intensity and in duration. There have been several papers in high impact factor journals dealing with the impacts of MHW in different compartments of marine ecosystems. This paper complements what has been done to date, extending the focus to the ecosystem structure and functioning. Therefore, the authors are to be congratulated for their work on this highly relevant topic.

Overall, the study is sound and interesting. However, the first doubt that I had while reading the paper was how illustrative this specific case study is from other MHW events in the area,

and in other areas. In this context, the study could benefit from some background information about the study area (including a map of the area), of the MHW specific event that is capturing and about the main changes that were recorded in terms of biomass and landings of different ecosystem elements. This information can provide the reader the background needed to understand how specific or general this case study is.

Another issue is the fact that MHWs are recurrent events that are characterized by a specific duration, intensity and extension (if they are only in the surface or also occurring at depth). A better description of the conditions in the area before, during and after the event is also needed to understand the type of MHW that this ecosystem went through. Also, a time series of MHW events in the area could be useful to understand if the one that has been studied in this particular study is frequent or had been a one-time event.

A third main comment is about the methodology. I understand that the authors have used a well-established modelling framework and methodology to quantitatively characterize the ecosystem before and after the MHW event. However, there is not a lot of information about the parameterization of both models, and in these decisions is the most important information in terms of methodology. Thus, further details about the parameters and decisions made to characterize both time periods is needed. For example, what is the type of flow control used for trophic interactions? How are metabolic rates parameterized according to different temperatures? In supplementary materials I see the same PB and QB parameters for both models. Is this realistic? What is the impact of this decision? What is the uncertainty related to these parameters and choices and how it has been included or could be included? Specifically, in the study the impact of temperature is mentioned (line 74). Is a change in temperature included in the models, how? And is this the only physical change included? In table 1, is it possible to add the SD of indicators?

Regarding methodology, it is not well explained why the chosen indicators are selected. More information about their sensitivity and specificity to extreme events is needed. In addition, I was wondering why a comparative approach using two static models has been chosen instead of a time dynamic or spatial-temporal dynamic approach. This is probably related to data accessibility and advantages and limitations of each approach. Some

explanation in the discussion about this could be useful.

The argument about the change of the ecosystem to a new state is difficult to follow. I would like to see in the discussion this argument complemented with additional literature about regime shifts and new states. The fact that some indicators are significantly different from one model to the other is not enough evidence to conclude that the whole system has changed. I suggest this part to be revisited.

Reviewer #3 (Remarks to the Author):

REVIEW

NCOMMS-445690_0

Title: Marine heatwaves disrupt ecosystem structure and function via altered food webs and energy flux

By Gomes et al.

GENERAL COMMENTS

The objective of this manuscript (ms) authored by Gomes et al. is to examine the impact of marine heatwaves (MHW) on ecosystem structure and function through the modification of food webs and energy flux. In order to achieve this objective, the authors conduct a comparative analysis of many outputs obtained from two end-to-end steady-state models, which accurately depict the mass balance of the NCC food web. These models are utilized to examine the changes that occurred in the food web before and after a major MHW event.

The authors concluded that an ecological regime shift occurred. Gelatinous taxa experienced the largest transformations, underscored by the arrival of northward-expanding pyrosomes. Despite altered trophic relationships and energy flux, the post-heatwave ecosystem appears stable, suggesting a shift to a novel ecosystem state.

Firstly, it is important to note that I am not an English native-speaker, as evident to any reader of this report. I will thus refrain from making any remarks concerning the English language of the ms. However, as far as I can tell, the manuscript is pleasant and easy to read, clear, and well-written.

The topic is of great importance, as it is now acknowledged that MHW are at least as influential in shaping biodiversity as global warming trends. The retained strategy is basic but effective: it compares the average organization and functioning of the food web during two distinct periods surrounding the region's primary MHW event. The present study is based on an update and adaptation of two previous and quite recent ECOPATH/ECOTRAN models already available in the literature. It has to be noted that the second one is only available on bioRxiv and that the "article is a preprint and has not been certified by peer review". At this stage, it's a limiting factor because I would have preferred to be able to rely on a basis validated by peers.

Ecotran is not as well-known, popular and widely-used as its bigger brother EwE. The scientific community thus probably lack significant backwardness and implementation expertise. Nevertheless, it is worth noting that both the pre-MHW Ecopath model and the post-MHW E2E Ecotran model exhibit a high degree of reliability, ecological plausibility, and adherence to the most appropriate standards in the field (Heymans et al., 2016 – doi: 10.1016/j.ecolmodel.2015.12.007). The comparison is valid and effectively executed.

Besides some specific comments listed below, there are, in my perspective, three main concerns to consider.

The use of an ECOTRAN approach is undoubtedly clever and relevant in the context of the NCC food web. Nevertheless, this is a mass-balanced / steady state approach. Thus, it does not account for dynamic adjustments and, by construction, provides a static and stable picture. I therefore believe that this raises a number of questions. First, according to the present ms and the cited research, at least one more heatwave occurred in 2019-2020, i.e. during what the authors referred to as the post-heatwave period (I68-69). This, in my perspective, might be extremely perplexing regarding the representativeness of the second

model. How far is it reasonable to consider that the second period might picture a potentially stable state? I suggest that perhaps the authors might be a little more precise about (1) the event(s) they want to consider and (2), if required, what they mean by post-heatwave. Furthermore, it is worth considering the plausibility that the ecological shift generated by MHWs may have resulted in variations in parameter values among the models. The lack of specification or discussion regarding this matter presents an intriguing opportunity for further exploration. For instance, quite recent papers by Maureaud et al. (2017) and du Pontavice et al. (2019, 2021) demonstrated the sensitivity of marine ecosystem trophodynamics to changes in ocean temperature (and in particular two metrics: the trophic transfer efficiency of energy through the food web and the residence time of biomass within each trophic level). In light of the authors' assessment of their relevance, how may these findings be incorporated into their deliberations for the current manuscript?

This brings us to the question of transfer efficiency or more exactly ecotrophic efficiency and food availability. Would examining EE values allow us to characterise trophic constraints during the second period? In fact, I find that the discussion of trophic limitation in the ms to be quite qualitative. Perhaps EEs might be used as an illustrative metric (with care obviously understanding how it is calculated in EwE.)

My last concern is about 'stability'. The authors concluded the ms arguing that "Ecosystem models may provide a means for determining whether disrupted marine ecosystems have entered an alternative stable state or are temporary instabilities, based on network metrics of stability and trophic connections." Although this statement is generally accurate, I don't think it is entirely persuasive in the context of this study. As previously stated, Ecopath/Ecotran inherently provide static representations of food webs in a steady-state condition. In my perspective, it is challenging to differentiate these images from stable conditions. Couldn't the conclusion appear a little tautological, if not incorrect, in that context? This observation holds particularly true for the second model, given the depicted time exhibits significant instability from an environmental perspective (as discussed above). Furthermore, part of the conclusion concerning stability in the food web functioning during the second period rely on connectance and link density values. It is advisable to exercise caution when utilising network metrics of this nature due to two primary reasons. Firstly,

the values of these metrics are contingent upon the level of aggregation employed in the model. Secondly, there are few, if ever, reference values available in the existing literature. Can I suggest that the authors at least be more nuanced in their conclusions?

Finally, my conclusion is that the present ms needs some revisions and edits in order to be considered for publishing in Nature Communications. However, I think it deserves to be published in the end given the relevance of the topic, the quality of the data, a high degree of reliability, ecological plausibility, and adherence to the most appropriate standards in the field of the models and the ecological interest of the conclusion.

SPECIFIC COMMENTS

(195-98) The assumption of a causal relationship (where less production availability for consumers results in a decrease in abundance) is enticing and likely significant. Nevertheless, I am curious in whether the used modelling approach enables a direct transition from "correlation" to causality, as discussed by Steele and Ruzicka (2011) in their work on ECOTRAN. This steady state solution formally describes the system in question and "does not determine cause and effect" which suggests that there is no ability to establish a causal relationship. Perhaps, for the sake of clarity and accuracy, the authors should be more nuanced in their statements. Could the decline in abundance of the mentioned species be attributed to environmental factors, similar to how the abundance of pyrosomes increased in response to marine heatwaves (MHW)?

(1107-108) Is this statement supported by any references? If so, what type of data (stomach contents, SIA, etc.) is it based on? In fact, and I'm sure the authors are aware of this as well as I am, stomach content analyses, for example, are not particularly relevant to identify consumption of gelatinous.

(1124-126) The authors suggest that the average trophic levels across a food web serve as a reliable indicator of the food chain length, which is the metric specifically addressed by Borrelli & Ginzburg (2014). Nevertheless, it is worth considering whether the association

between chain length and average trophic level remains unequivocal, particularly when the estimation of trophic levels relies on the utilization of an ECOPATH modeling framework. It appears that the estimation of TL (trophic level) in ECOPATH is partially correlated with the level of aggregation of trophic groups. Furthermore, in the context of ECOTRAN, where the practice of auto-consumption is practically not allowed.

(1138-140) I guess that the authors wanted to say that the post-MHW food web MAY PROBABLY BE MORE stable. I do find it challenging to draw conclusions about stability using steady-state pictures.

(1269) I may have overlooked a crucial aspect, but I failed to comprehend the amount by which uncertainty is calibrated, particularly with regard to the specific amount of uncertainty assigned to parameter values. Was each matrix element varied randomly within $\pm 50\%$ from a specific distribution (presumably a normal distribution, in the present case) as in the original Ruzicka et al. model? If that is the case, it is necessary to address the significance of the 50% value. For example, Christensen et al. acknowledge that more than 80% of uncertainty exists for diet matrix elements. By the way, if I understand well, as in previous studies, the authors have focused on addressing uncertainty specifically in the element of the production matrix, rather than considering uncertainty across all parameters. Although this approach differs from other uncertainty approaches employed in ECOPATH, such as the ENA-tool (Guesnet et al., 2015 – doi: 10.1016/j.ecolmodel.2015.05.036) and Ecosampler (Steenbek et al., 2018 – doi: 10.1016/j.softx.2018.06.004), which encompass uncertainty in a broader range of parameters, it's pragmatic and in some ways quite elegant as it is presumed to comprehensively address uncertainty across many hierarchical levels.

References

du Pontavice, H., Gascuel, D., Reygondeau, G., Maureaud, A., & Cheung, W. W. L. (2019). Climate change undermines the global functioning of marine food webs. *Global Change Biology*, 26(3), 1306–1318. <https://doi.org/10.1111/gcb.14944>

du Pontavice, H., Gascuel, D., Reygondeau, G., Stock, C., & Cheung, W. W. L. (2021). Climate-

induced decrease in biomass flow in marine food webs may severely affect predators and ecosystem production. *Global Change Biology*, gcb.15576.

<https://doi.org/10.1111/gcb.15576>

Maureaud, A., Gascuel, D., Colléter, M., Palomares, M. L. D., Pontavice, H. D., Pauly, D., & Cheung, W. W. L. (2017). Global change in the trophic functioning of marine food webs.

PLOS ONE, 12(8), e0182826. <https://doi.org/10.1371/JOURNAL.PONE.0182826>

REVIEWER COMMENTS (responses in blue)

Reviewer #1 (Remarks to the Author):

This manuscript uses modelling to explore the ecosystem-level impacts of persistent marine heatwaves in the Northeast Pacific Ocean. More specifically, the authors explore the differences between two Ecotran models fit to data from the region pre- and post- onset of the marine heatwaves to evaluate changes in the structure of the ecosystem in the Northeast Pacific Ocean in response to heatwaves. They find noticeable changes in energy flow and composition of the ecosystem pre- and post- heatwave. I thought the paper was easy to understand and interesting, exploring a topic of likely significant interest to a wide audience. I have one significant comment, and then some minor comments about presentation.

Significant comment:

From what I understand, the structure of the ecosystems as represented in the Ecotran models are static, since they are fit to data from the system they're representing (Northeast Pacific Ocean pre- and post- heatwave). This means that the primary linkage between the pre- and post- heatwave models are through the data; the post- heatwave model configuration does not emerge from the pre- heatwave model, in response to changes in environmental conditions. To me, this is a significant limitation that should be discussed in the main text, because I think it makes the arguments around novel regime shifts, and the stability of the post-MHW system unconvincing (lines 121-140). The pre-MHW system changed in response to shifting environmental conditions, to a post-MHW structure that has similar metrics of stability as the pre-MHW system (Table 1). But, what's stopping the post-MHW system (in the real world, not in the model) from moving to an alternative state again, in response to even greater warming in the future? If the pre-MHW changed, even though it wasn't unstable, why wouldn't the post-MHW system change again? This is a significant limitation of the modelling approach I think (unless I'm missing something); the model ecosystem structure is baked in, it can't change in response to environmental conditions, even though the real system changed and may (will?) change again. The post-MHW model does not emerge from the pre-MHW configuration, the only link between the two is that the observations come from the same system. This doesn't mean the results aren't interesting, or the comparison isn't sound, but it means that I don't think you can talk about the stability of regime shifts convincingly, since an already stable system (by your metrics in Table 1) shifted in response to environmental conditions.

Thank you for this comment. We agree that the pre-MHW ecosystem shifted, but this doesn't mean it was necessarily unstable prior to the perturbation (i.e., stable without a perturbation to push it to another state). We agree that the new ecosystem may yet shift again (regardless of whether or not it is stable), following a large perturbation to the ecosystem. Yet, assuming no further perturbations to a system it is possible for the ecosystem to either i) revert to initial conditions (if the new system is not stable) or ii) stay in this new state (if the new state is stable), again assuming no further dramatic perturbations occur (which we cannot predict).

We disagree that having the post-MHW model emerge from the pre-MHW model (rather than two independently parameterized, but otherwise similar models) makes it any better of a method for understanding ecosystem stability or regime shifts. Each of these methods would have advantages and disadvantages (e.g., computation time, model assumptions, parameter tuning,

etc.; see also responses to Reviewer 2), but there is no reason to believe that one method can assess this question better than the other.

With this said, we agree with your assessment that we should be more cautious in how we discuss ecosystem stability and regime shifts, especially given our use of steady-state models. Thus, we have removed all interpretation and language around ecosystem stability and regime shifts in both the abstract and main text to avoid contention.

Minor comments:

Line 153 - the reference to Figure 1 to discuss the increased dominance of jack mackerel is almost impossible to find in the Figure. I suggest highlighting the group differently, or finding another way to reference the change.

Thank you for this comment. We have now bolded the numbers in Figure 1b (#35 for jack mackerel) as we have done in Figure 1c, so that the group can more easily be located.

Line 254-259 - footprint/reach sounds like the same thing to me, from what's written. An example of each would really help.

Thank you for this comment. We have now added two additional sentences and further language clarifying the confusing definitions of footprint and reach. The paragraph now reads:

“A functional group’s trophic impact upon lower trophic levels is expressed by its footprint, which is the fraction of each producer group’s total production supporting focal consumer groups via all direct and indirect pathways (excluding detritus). The footprint, in other words, is the proportion of energy from lower trophic levels (relative to a focal group) consumed by that focal group. Conversely, the importance of a focal functional group to higher trophic levels was expressed by its reach: the fraction of consumers’ production that originated with (or passed through) that functional group via all direct and indirect pathways³⁷. Thus, the reach is the proportion of energy consumed by higher trophic levels (relative to a focal group) that passed through that functional group.”

Line 269 - what is a "standard" values for the CV?

Thank you for this good question. Our CV values are based on the standard Ecopath pedigree strategy of assigning uncertainty based on the types of (i.e., survey type) and confidence in (quantity and quality) our data sources. In our case, "standard" meant that we used a common or generic set of conservative uncertainties for all trophic interactions. We’ve added language clarifying this in the manuscript as well as pointed readers to where they can find these values in the supplemental data and code repository:

“The net uncertainty among physiological parameters, diet, and nutrient cycling terms are expressed as levels of uncertainty about each element of the production matrix. In our analyses, each element of the production matrix was randomly varied by drawing model parameters from a normal distribution with a mean of the originally parameterized value and a standard deviation [converted from conservative coefficient of variation (CV)

values, which were based on the Ecopath pedigree strategy of assigning uncertainty based on the types of (i.e., survey type) and confidence in (quantity and quality) our data sources; see supplemental data and code repository readme]. For example, we used diet information from 39083 individual juvenile Chinook salmon, 2911 Pacific herring, and seven pyrosome colonies³⁶. Thus, we set CV values for each element of their diet vectors to 0.1, 0.5, and 0.8, respectively to reflect differences in the robustness of the datasets (and the associated uncertainty). We drew 1000 possible Monte Carlo food web models to investigate the propagation of uncertainty for footprint and reach of each assessed functional group within each model (pre-MHW and MHW).”

Figure 1 - what is the unit for "the magnitude of the change in biomass", i.e., the size of the circles? A reference circle is needed for the delta plot.

Thank you for this question and comment. The magnitude is unitless, as it is the relative change (multiplication factor) to go from one state to the other. We have now added a scale bar with the corresponding multiplication factors to the figure. We agree that this makes the figure much more readable.

Figure 2b - y-axis label needs to be more informative. When I see "proportion" I ask why it doesn't go from 0 to 1. It's explained in the figure caption but it should be easy to understand from the figure itself.

Thank you for this point. We have now added a short description to the y-axis label to reflect that the non-living groups are not shown and have added additional clarification in the figure caption. We hope that this clarifies any confusion about these data being proportions.

Figure 3 - what are the units for "ecosystem-wide consumption and contribution values for representative taxa"? Are these percentages?

Thank you for this question. We have now clarified that these are proportions in the figure axis and figure legend.

Reviewer #2 (Remarks to the Author):

This study aims at quantitatively describing changes in the structure of a marine ecosystem due to a marine heatwave (MHW). The topic is highly relevant since MHW are increasing in frequency, in intensity and in duration. There have been several papers in high impact factor journals dealing with the impacts of MHW in different compartments of marine ecosystems. This paper complements what has been done to date, extending the focus to the ecosystem structure and functioning. Therefore, the authors are to be congratulated for their work on this highly relevant topic.

Thank you for the positive feedback.

Overall, the study is sound and interesting. However, the first doubt that I had while reading the

paper was how illustrative this specific case study is from other MHW events in the area, and in other areas. In this context, the study could benefit from some background information about the study area (including a map of the area), of the MHW specific event that is capturing and about the main changes that were recorded in terms of biomass and landings of different ecosystem elements. This information can provide the reader the background needed to understand how specific or general this case study is.

Thank you for these comments. We were not sufficiently clear in our initial manuscript version that we are referring to multiple successive MHW events (rather than a single MHW as a case study). Multiple recurring MHWs have kept the NCC in a state of anomalously warm conditions since late 2013. We've added language to clarify this throughout, including changing all text referencing the post-MHW model to now be named the 'MHW' model to better reflect the time period during which we are referring to. We also have added a time series to Figure 1, which includes a depiction of what time periods we are capturing with our modeling and how many years the system has experienced anomalously high SSTs. We have added additional details of some specific changes that were recorded during the early MHWs within the third paragraph of the manuscript, per your suggestions, but we note that there are many changes that have occurred throughout the multiple North Pacific Ocean MHWs that are difficult to detail comprehensively here (due to space limitations). Thus, in addition to the new text, we provide references that supply these details (including maps) to encourage the reader to further learn about the previously documented physical and biological changes elsewhere. We also provide a table (Table S1) that includes biomass changes of each functional group between the two ecosystem model states.

Another issue is the fact that MHWs are recurrent events that are characterized by a specific duration, intensity and extension (if they are only in the surface or also occurring at depth). A better description of the conditions in the area before, during and after the event is also needed to understand the type of MHW that this ecosystem went through. Also, a time series of MHW events in the area could be useful to understand if the one that has been studied in this particular study is frequent or had been a one-time event.

Thank you for these comments. Similar to the comment above, we now clarify that we are referring to multiple recurring MHWs that have kept the NCC in a state of anomalously warm conditions since late 2013. We have added clarifying language to the manuscript, including changing all text referencing the post-MHW model to now be named the 'MHW' model to better reflect that we are referring to during the MHW period. We also have added a time series of SST anomalies to Figure 1, and have further characterized the MHWs that the NCC has experienced over the past decade, and point the readers to references where they can learn more about these specific MHWs from studies published by physical oceanographers who provide much more detail on the duration, intensity, and characteristics of SST anomalies over time. We agree that these additional details have improved the quality of our manuscript.

A third main comment is about the methodology. I understand that the authors have used a well-established modelling framework and methodology to quantitatively characterize the

ecosystem before and after the MHW event. However, there is not a lot of information about the parameterization of both models, and in these decisions is the most important information in terms of methodology. Thus, further details about the parameters and decisions made to characterize both time periods is needed. For example, what is the type of flow control used for trophic interactions? How are metabolic rates parameterized according to different temperatures? In supplementary materials I see the same PB and QB parameters for both models. Is this realistic? What is the impact of this decision? What is the uncertainty related to these parameters and choices and how it has been included or could be included? Specifically, in the study the impact of temperature is mentioned (line 74). Is a change in temperature included in the models, how? And is this the only physical change included? In table 1, is it possible to add the SD of indicators?

Thank you for these comments. We have now added details to the Methodology surrounding the model-building process and the uncertainty of model parameters (see responses above for added uncertainty text and tracked changes manuscript file). With this said, we do not want to be overly verbose in our description here since previous work (Gomes et al. 2022 and Ruzicka et al. 2012, both references found in the manuscript) outlines many of these details there. We do not believe we have the space in such a short form article such as Nature Communications to provide all of these details, but agree that the further clarifying details that we have added here are useful.

You are correct that the physiology parameters are the same between the two different models, which is more of a limitation of available information within the ecosystem than an active choice. The MHWs are recent enough that we do not have data on how these physiology parameters (e.g., growth and consumption) might be different before and after the onset of the MHWs in this ecosystem. While it is likely that these parameters do differ for some groups, we are not comfortable making assumptions about which of these groups likely differ. We believe that keeping these parameters the same across models offers a more conservative approach to understanding ecosystem differences. The fact that these physiology parameters also differ (in combination with the biomass and diet differences already incorporated into this analysis), likely only exacerbates the findings that the ecosystem differs since the onset of MHWs. With this said, we have now added some calculations to the discussion (and methods) about what a 2 degree C change in water temperatures might mean for physiology of marine organisms. We selected 2 degrees because temperatures in the California Current exceeded a change in 2 degrees during both major MHWs (2013/2014 and 2019/2020), and it is a commonly referenced hypothetical change in climate scenarios. This may simply be a difference in terminology, but we are unsure of what you mean by 'flow control.'

Temperature is not explicitly included in this model, but instead is indirectly included through the inclusion of a MHW period – that is, we are comparing ecosystem models between the two different states (pre- and post-onset of MHWs). Processes will be included in future implementations of EcoTran through step-by-step improvement – inclusion of water temperature, and the physiological relationships of temperature across various functional groups are in active development, but beyond the scope of this current work.

We have now added the standard deviations to Table 1 per your request.

Regarding methodology, it is not well explained why the chosen indicators are selected. More information about their sensitivity and specificity to extreme events is needed. In addition, I was wondering why a comparative approach using two static models has been chosen instead of a time dynamic or spatial-temporal dynamic approach. This is probably related to data accessibility and advantages and limitations of each approach. Some explanation in the discussion about this could be useful.

Thank you for these comments. We chose to display information that highlight the differences in energy flow pathways between the two ecosystem states (Figures 1 and 2) and to highlight differences among functional groups in terms of their energy demands and the contributions they provide to higher trophic levels (Figure 3). The later, footprint and reach, have been used extensively in previous EcoTran work (see Ruzicka et al. references below). Table 1 shows common network metrics that are used either descriptively or in some comparative context. None of these comparisons (Figures 1-3, Table 1) are particularly specific to 'extreme events' as they can be used to compare between any set of ecosystem models (parameterized for different years, locations, etc.). Yet, there is nothing inherently different about analyzing ecosystem models that are subjectively separated by so called extreme events. That is, our modeling and analysis – comparisons across ecosystem models – is not any less viable for prolonged MHWs than it is for any other difference one might be interested in (years, locations, pre- and post-management interventions etc.).

For network metrics (Table 1) and for footprint and reach metrics (Figure 3), our Monte Carlo analysis generated 1000 variations of the food web that allowed us to evaluate the net uncertainty among all parameters that define the strength of each trophic relation (e.g., biomass, consumption rates, growth efficiency, etc.). That is, the overall sensitivity of model-derived metrics to parameter uncertainty is plotted as the spread about the median values in the Fig. 3 violin plots or as standard deviations about the mean in Table 1. This is perhaps not what you are referring to as sensitivity as we did not evaluate the role of each parameter, individually, but it does provide a measure of overall uncertainty in our derived metrics. We do not believe that these metrics would be any more or less sensitive to ecosystem models parameterized during extreme events.

A comparative approach using two static models has several advantages. Firstly, the static models represent the mean ecosystem states over distinct but prolonged periods of time (months to years). To make meaningful inferences of ecosystem state, they do not require resolution on short time scales of physiological responses to changing environmental conditions such as the relationships of growth, consumption, nutrient and detritus recycling with temperature. By leveraging existing datasets, we are able to capture two distinct snapshots of what the ecosystem looks like both before and after the onset of marine heatwaves. Physiological responses with temperature are actively being incorporated into time-dynamic EcoTran simulations, but this work still requires considerable effort, and is outside the scope of this manuscript. Relatedly, the static comparison approach requires fewer assumptions about those relationships and whether they are captured adequately. Lastly, static models are computationally quicker to run and interpret. Time-dynamic counterparts require much more iterative tuning to make sure that the temporal dynamics make sense before models can be

interpreted temporally. Static models serve as a simplified snapshot, but useful assessment of ecosystem differences.

Ruzicka, J. J., Daly, E. A. & Brodeur, R. D. Evidence that summer jellyfish blooms impact Pacific Northwest salmon production. *Ecosphere* 7, e01324 (2016).

Ruzicka, J. J. *et al.* Interannual variability in the Northern California Current food web structure: changes in energy flow pathways and the role of forage fish, euphausiids, and jellyfish. *Progress in Oceanography* 102, 19–41 (2012).

The argument about the change of the ecosystem to a new state is difficult to follow. I would like to see in the discussion this argument complemented with additional literature about regime shifts and new states. The fact that some indicators are significantly different from one model to the other is not enough evidence to conclude that the whole system has changed. I suggest this part to be revisited.

Thank you for these comments. After careful consideration of your comments, along with the comments of the other two reviewers, we agree with the general sentiment that our analyses do not contain enough evidence to conclusively determine that the ecosystem has undergone a regime shift and entered a novel stable state. Thus, we have removed all language suggesting that this is the case. Thank you for flagging this point.

Reviewer #3 (Remarks to the Author):

REVIEW

NCOMMS-445690_0

Title: Marine heatwaves disrupt ecosystem structure and function via altered food webs and energy flux
By Gomes et al.

GENERAL COMMENTS

The objective of this manuscript (ms) authored by Gomes et al. is to examine the impact of marine heatwaves (MHW) on ecosystem structure and function through the modification of food webs and energy flux. In order to achieve this objective, the authors conduct a comparative analysis of many outputs obtained from two end-to-end steady-state models, which accurately depict the mass balance of the NCC food web. These models are utilized to examine the changes that occurred in the food web before and after a major MHW event.

The authors concluded that an ecological regime shift occurred. Gelatinous taxa experienced the largest transformations, underscored by the arrival of northward-expanding pyrosomes. Despite altered trophic relationships and energy flux, the post-heatwave ecosystem appears stable, suggesting a shift to a novel ecosystem state.

Firstly, it is important to note that I am not an English native-speaker, as evident to any reader of this report. I will thus refrain from making any remarks concerning the English language of the ms. However, as far as I can tell, the manuscript is pleasant and easy to read, clear, and well-written.

Thank you for the positive feedback.

The topic is of great importance, as it is now acknowledged that MHW are at least as influential in shaping biodiversity as global warming trends. The retained strategy is basic but effective: it compares the average organization and functioning of the food web during two distinct periods surrounding the region's primary MHW event. The present study is based on an update and adaptation of two previous and quite recent ECOPATH/ECOTRAN models already available in the literature. It has to be noted that the second one is only available on bioRxiv and that the "article is a preprint and has not been certified by peer review". At this stage, it's a limiting factor because I would have preferred to be able to rely on a basis validated by peers.

Thank you for bringing this up. Indeed Gomes et al. 2022 is still a pre-print, however it is in the late stages of peer-review at PLOS ONE. Unfortunately, there have been considerable delays in receiving reviews (86 and 111 days awaiting reviewer comments in rounds 1 and 2, respectively, which is representative of our general experience with the peer review process since the start of the covid-19 pandemic), that has delayed this manuscript being officially considered "certified by peer review." However, in the latest round of reviews, which were recently submitted, there were very few issues to address and, thus, we hope that it will be at least accepted before the present manuscript goes to press. The bioRxiv pre-print has been updated to reflect the most recent versions resubmitted to address reviewer comments, and those updates are reflected in the analyses used for the present manuscript.

Ecotran is not as well-known, popular and widely-used as its bigger brother EwE. The scientific community thus probably lack significant backwardness and implementation expertise. Nevertheless, it is worth noting that both the pre-MHW Ecopath model and the post-MHW E2E Ecotran model exhibit a high degree of reliability, ecological plausibility, and adherence to the most appropriate standards in the field (Heymans et al., 2016 – doi: 10.1016/j.ecolmodel.2015.12.007). The comparison is valid and effectively executed.

Thank you for the positive feedback!

Besides some specific comments listed below, there are, in my perspective, three main concerns to consider.

The use of an ECOTRAN approach is undoubtedly clever and relevant in the context of the NCC food web. Nevertheless, this is a mass-balanced / steady state approach. Thus, it does not account for dynamic adjustments and, by construction, provides a static and stable picture. I therefore believe that this raises a number of questions. First, according to the present ms and the cited research, at least one more heatwave occurred in 2019-2020, i.e. during what the authors referred to as the post-heatwave period (l68-69). This, in my perspective, might be extremely perplexing regarding the representativeness of the second model. How far is it reasonable to consider that the second period might picture a potentially stable state? I suggest

that perhaps the authors might be a little more precise about (1) the event(s) they want to consider and (2), if required, what they mean by post-heatwave.

Thank you for these points. We have now clarified further that we are considering the period from the start of the winter 2013/2014 MHW through 2022, including the 2019-2020 heatwave, and all the other years with anomalously high SSTs (see new Figure 1a and introductory text). We've also now clarified that we mean post-onset of the initial marine heatwave, rather than post-heatwave, and have, thus, renamed 'post-MHW' to 'MHW'. We hope that this has clarified any confusion that our wording led to. Additionally, we have removed language around stable states to avoid any contention about whether or not our analysis adequately assesses ecosystem stability (see responses to other reviewers as well).

Furthermore, it is worth considering the plausibility that the ecological shift generated by MHWs may have resulted in variations in parameter values among the models. The lack of specification or discussion regarding this matter presents an intriguing opportunity for further exploration. For instance, quite recent papers by Maureaud et al. (2017) and du Pontavice et al. (2019, 2021) demonstrated the sensitivity of marine ecosystem trophodynamics to changes in ocean temperature (and in particular two metrics: the trophic transfer efficiency of energy through the food web and the residence time of biomass within each trophic level). In light of the authors' assessment of their relevance, how may these findings be incorporated into their deliberations for the current manuscript?

Thank you for these points. We agree that both biomass residence time and trophic transfer efficiency changes due to ocean temperatures present an intriguing opportunity for further exploration. Thus we have added an additional paragraph (second to last in the main text) evaluating the hypothetical effects that a change in 2 C would have in terms of changes to P/B rate and in terms of changes in metabolic rates and production efficiency. To support this paragraph, we have also added an additional section to the Methods titled "Estimates of biomass flow and trophic transfer efficiency changes due to temperature" where we outline our calculations following the references you've supplied and those therein.

This brings us to the question of transfer efficiency or more exactly ecotrophic efficiency and food availability. Would examining EE values allow us to characterise trophic constraints during the second period? In fact, I find that the discussion of trophic limitation in the ms to be quite qualitative. Perhaps EEs might be used as an illustrative metric (with care obviously understanding how it is calculated in EwE.)

Thank you for this point. EE values between the two models differ very little in most cases (see Table S1). In our experience, these values are mainly used as mass-balancing, or tuning, parameters, and they aren't very reliable as precise indicators of trophic limitation. With that said, to explore how increased temperatures might alter EE values, we apply a multiplication factor (see new Methods section titled "Estimates of biomass flow and trophic transfer efficiency changes due to temperature"), of 1.036656 for our hypothetical increased temperature (2°C) production to biomass ratios, and we see an average reduction in EE values of 0.0048 going from pre-MHW to a 2°C warmer model, with the largest change being a difference of 0.029 (tunas). In our experience, changes of this magnitude in EE values are negligible. That is, these

changes can occur from very minor changes to biomass or diet estimates that are well within the conservative uncertainty bounds we have set for those parameters. We also considered the effect that a 2°C would have on the metabolic rates, production efficiencies, and transfer efficiencies of poikilothermic groups.

My last concern is about 'stability'. The authors concluded the ms arguing that "Ecosystem models may provide a means for determining whether disrupted marine ecosystems have entered an alternative stable state or are temporary instabilities, based on network metrics of stability and trophic connections." Although this statement is generally accurate, I don't think it is entirely persuasive in the context of this study. As previously stated, Ecopath/Ecotran inherently provide static representations of food webs in a steady-state condition. In my perspective, it is challenging to differentiate these images from stable conditions. Couldn't the conclusion appear a little tautological, if not incorrect, in that context? This observation holds particularly true for the second model, given the depicted time exhibits significant instability from an environmental perspective (as discussed above). Furthermore, part of the conclusion concerning stability in the food web functioning during the second period rely on connectance and link density values. It is advisable to exercise caution when utilising network metrics of this nature due to two primary reasons. Firstly, the values of these metrics are contingent upon the level of aggregation employed in the model. Secondly, there are few, if ever, reference values available in the existing literature. Can I suggest that the authors at least be more nuanced in their conclusions?

Thank you for these points, they are well taken. We agree that representing the ecosystem with steady-state models and then assessing their stability via independent (i.e., not baked-in to EcoTran) metrics can appear tautological or circular. For this reason, and for the other points of caution around stability that you and the other reviewers have raised, we have decided to be much more cautious in our manuscript and have removed any indication that we have adequately measured ecosystem stability or regime shifts.

While moot at this point, we disagree that the second model should show significant instability. While many functional groups increased or decreased (even to the point of mass mortality events) during the MHWs, this doesn't mean that the ecosystem has not entered (or is entering) a new state that is stable following a series of significant perturbations.

We also agree that the use of network metrics is contingent upon the level of aggregation, however both networks were built upon the same levels of aggregation. Thus, relative differences between the two must be based upon differences in the linkages between nodes, since the nodes are the same. This comparison of relative values should not necessitate absolute reference values to be available in the literature, since we are making a relative (not absolute) comparison. With this said, we agree with your earlier points about tautological argumentation, and have removed language around ecosystem stability.

Finally, my conclusion is that the present ms needs some revisions and edits in order to be considered for publishing in Nature Communications. However, I think it deserves to be published in the end given the relevance of the topic, the quality of the data, a high degree of reliability, ecological plausibility, and adherence to the most appropriate standards in the field of

the models and the ecological interest of the conclusion.

Thank you for the positive feedback!

SPECIFIC COMMENTS

(I95-98) The assumption of a causal relationship (where less production availability for consumers results in a decrease in abundance) is enticing and likely significant. Nevertheless, I am curious in whether the used modelling approach enables a direct transition from "correlation" to causality, as discussed by Steele and Ruzicka (2011) in their work on ECOTRAN. This steady state solution formally describes the system in question and "does not determine cause and effect" which suggests that there is no ability to establish a causal relationship. Perhaps, for the sake of clarity and accuracy, the authors should be more nuanced in their statements. Could the decline in abundance of the mentioned species be attributed to environmental factors, similar to how the abundance of pyrosomes increased in response to marine heatwaves (MHW)?

Thank you for this point. This is well taken. We have added "may have" and "possibly" to these two sentences, as well as changed "led to" to "contributed to" to more accurately reflect our uncertainty about the causality of these processes. Yes, you are absolutely right that we cannot disentangle direct and indirect (via the foodweb) environmental factors.

(I107-108) Is this statement supported by any references? If so, what type of data (stomach contents, SIA, etc.) is it based on? In fact, and I'm sure the authors are aware of this as well as I am, stomach content analyses, for example, are not particularly relevant to identify consumption of gelatinous.

Thank you for this point. Yes, this is based on recent stomach content data in the region (we have now included a few references here). Given this information, it appears that pyrosomes are not consumed as readily as jellies. We agree that stomach content analysis is particularly poor at detecting gelatinous material, but jellies are much softer bodied than pyrosomes. Although pyrosomes are considered gelatinous, they are not as compressible as soft-bodied jellies and it is likely that only the largest (mostly non-fish) predators can consume full size pyrosomes. So despite the fact that jellies are almost certainly more likely to be destroyed before detection (relative to pyrosomes), they still show up more consistently in diets. Brodeur et al. (2019, 2021) on the shift of demersal and pelagic predators to gelatinous prey in 2015 and 2016 showed that most of these prey are salps and no pyrosomes were consumed by any of the forage fishes despite being very numerous in the plankton.

Brodeur, R. D., Hunsicker, M. E., Hann, A., & Miller, T. W. (2019). Effects of warming ocean conditions on feeding ecology of small pelagic fishes in a coastal upwelling ecosystem: a shift to gelatinous food sources. *Marine Ecology Progress Series*, 617, 149-163.

Brodeur, R. D., Buckley, T. W., Lang, G. M., Draper, D. L., Buchanan, J. C., & Hibpshman, R. E. (2021). Demersal fish predators of gelatinous zooplankton in the Northeast Pacific Ocean. *Marine Ecology Progress Series*, 658, 89-104.

(1124-126) The authors suggest that the average trophic levels across a food web serve as a reliable indicator of the food chain length, which is the metric specifically addressed by Borrelli & Ginzburg (2014). Nevertheless, it is worth considering whether the association between chain length and average trophic level remains unequivocal, particularly when the estimation of trophic levels relies on the utilization of an ECOPATH modeling framework. It appears that the estimation of TL (trophic level) in ECOPATH is partially correlated with the level of aggregation of trophic groups. Furthermore, in the context of ECOTRAN, where the practice of auto-consumption is practically not allowed.

Thank you for this point. Yes Borrelli & Ginzburg (2014) address food chain length, but, per their definition, this is inextricably linked to trophic levels “Food chain length is directly related to the number of trophic levels in a food web. A food chain with two steps has three trophic levels...” (Borrelli & Ginzburg, 2014). With that said, we agree that the maximum chain length and average trophic level is not necessarily the same thing. We’ve altered the language around this to better reflect the Borrelli and Ginzburg reference. We agree that TL estimation is contingent upon the aggregation of trophic groups, however, in our model comparison trophic groups are defined the same way such that average TL estimation differences should only be based on differences in biomass and diets between the two models.

(1138-140) I guess that the authors wanted to say that the post-MHW food web MAY PROBABLY BE MORE stable. I do find it challenging to draw conclusions about stability using steady-state pictures.

Thank you for this point. We agree with your point, and we have now removed language around ecosystem stability per your suggestions here and elsewhere.

(1269) I may have overlooked a crucial aspect, but I failed to comprehend the amount by which uncertainty is calibrated, particularly with regard to the specific amount of uncertainty assigned to parameter values. Was each matrix element varied randomly within $\pm 50\%$ from a specific distribution (presumably a normal distribution, in the present case) as in the original Ruzicka et al. model? If that is the case, it is necessary to address the significance of the 50% value. For example, Christensen et al. acknowledge that more than 80% of uncertainty exists for diet matrix elements. By the way, if I understand well, as in previous studies, the authors have focused on addressing uncertainty specifically in the element of the production matrix, rather than considering uncertainty across all parameters. Although this approach differs from other uncertainty approaches employed in ECOPATH, such as the ENA-tool (Guesnet et al., 2015 – doi: 10.1016/j.ecolmodel.2015.05.036) and Ecosampler (Steenbek et al., 2018 – doi: 10.1016/j.softx.2018.06.004), which encompass uncertainty in a broader range of parameters, it's pragmatic and in some ways quite elegant as it is presumed to comprehensively address uncertainty across many hierarchical levels.

Thank you for this point. Our CV values are based on the standard Ecopath pedigree strategy of assigning uncertainty based on the types of (i.e., survey type) and confidence in (quantity and quality) our data sources. We used a common or generic set of conservative uncertainties for all trophic interactions. We’ve added language clarifying this in the manuscript as well as pointed readers to where they can find these values in the supplemental data and code repository.

While you are correct that we have addressed uncertainty in the production matrix, we also point out that we have considered uncertainty across all parameters. The uncertainty for each element of the production matrix is a product of uncertainties for each parameter contributing to the production matrix (diet, physiology, biomass). Thus, we did not select one value (i.e., 50%) with which to assess uncertainty, but instead made informed decisions about the quality and quantity of available data. For example, we used diet information from nearly 40,000 juvenile Chinook salmon, roughly 3,000 Pacific herring, and seven pyrosome colonies. Thus, we set CV values for each element of their diets to 0.1, 0.5, and 0.8, respectively to reflect differences in the robustness of the datasets (and the associated uncertainty). We have updated the Methods to reflect these details, and agree that they make the methodology more transparent.

References

- du Pontavice, H., Gascuel, D., Reygondeau, G., Maureaud, A., & Cheung, W. W. L. (2019). Climate change undermines the global functioning of marine food webs. *Global Change Biology*, 26(3), 1306–1318. <https://doi.org/10.1111/gcb.14944>
- du Pontavice, H., Gascuel, D., Reygondeau, G., Stock, C., & Cheung, W. W. L. (2021). Climate-induced decrease in biomass flow in marine food webs may severely affect predators and ecosystem production. *Global Change Biology*, gcb.15576. <https://doi.org/10.1111/gcb.15576>
- Maureaud, A., Gascuel, D., Colléter, M., Palomares, M. L. D., Pontavice, H. D., Pauly, D., & Cheung, W. W. L. (2017). Global change in the trophic functioning of marine food webs. *PLOS ONE*, 12(8), e0182826. <https://doi.org/10.1371/JOURNAL.PONE.0182826>

REVIEWERS' COMMENTS

Reviewer #1 (Remarks to the Author):

Thank you for your responses to my queries. I am satisfied with the changes you have made and recommend publication.

Reviewer #2 (Remarks to the Author):

This study aims at quantitatively describing changes in the structure of a marine ecosystem due to a marine heatwave (MHW). The topic is highly relevant since MHW are increasing in frequency, in intensity and in duration. There have been several papers in high impact factor journals dealing with the impacts of MHW in different compartments of marine ecosystems. This paper complements what has been done to date, extending the focus to the ecosystem structure and functioning. Therefore, the authors are to be congratulated for their work on this highly relevant topic. Overall, the study is sound and interesting.

Previous comments to the study have been addressed correctly and I am confident that main limitations of the study have been mentioned and addressed.

Reviewer #3 (Remarks to the Author):

Gomes et al. submitted here a revised version of their ms titled "Marine heatwaves disrupt ecosystem structure and function via altered food webs and energy flux" in this journal. The writers greatly enhanced their ms and effectively addressed the majority of the reviewers' issues. I congratulate the authors for taking the time and attention to consider the feedback from the reviewers. I'd also like to thank the other reviewers for their thoughtful and significant recommendations. The material appears to be much clearer now in its both purposes and conclusions. Especially considering the characterization of the two periods and the debate on stability.

Regarding this first point, I find that the objective of the paper is now clearer. There is no longer any ambiguity where the previous wording might have suggested that the objective

was to see how limited episodes of MHW could have impacted the food web. It is now clear that the aim is to compare two periods, a reference period before the first, historical MHW episode and a prolonged period of thermal anomalies marked by at least 2 referenced MHWs (or Blobs). On this point, however, I think the authors could have been even clearer, for example at the end of page 4, by specifying that the recent period is a prolonged period of thermal anomalies marked by at least 2 referenced MHWs. A distracted reader might think that the whole period (8 years) is considered a single episode of MHW, but that's not how it's described in the literature. I feel it is all the more necessary to insist, otherwise the name of the period (MHW period) could once again lead to confusion.

Another small but revealing detail, p5, following my prior comments, the authors notably reworded the section on TL, acknowledging in their response that average TL is not so directly related to chain length. However, there is still some ambiguity because the measure they used is average TL and the conclusion they draw is about the number of trophic levels. Both measures are still not identical. I admit that it isn't particularly important to the subject, but I believe it is always best to be exact in what you say.

Apart from these details, I think that the authors have answered my questions and comments in a relevant and well-argued manner. I hope the other reviewers will be as satisfied as I was, because I think this work deserves to be published.

REVIEWERS' COMMENTS

Reviewer #1 (Remarks to the Author):

Thank you for your responses to my queries. I am satisfied with the changes you have made and recommend publication.

Thank you for your help in improving the manuscript.

Reviewer #2 (Remarks to the Author):

This study aims at quantitatively describing changes in the structure of a marine ecosystem due to a marine heatwave (MHW). The topic is highly relevant since MHW are increasing in frequency, in intensity and in duration. There have been several papers in high impact factor journals dealing with the impacts of MHW in different compartments of marine ecosystems. This paper complements what has been done to date, extending the focus to the ecosystem structure and functioning. Therefore, the authors are to be congratulated for their work on this highly relevant topic. Overall, the study is sound and interesting.

Previous comments to the study have been addressed correctly and I am confident that main limitations of the study have been mentioned and addressed.

Thank you for your help in improving the manuscript.

Reviewer #3 (Remarks to the Author):

Gomes et al. submitted here a revised version of their ms titled "Marine heatwaves disrupt ecosystem structure and function via altered food webs and energy flux" in this journal. The writers greatly enhanced their ms and effectively addressed the majority of the reviewers' issues. I congratulate the authors for taking the time and attention to consider the feedback from the reviewers. I'd also like to thank the other reviewers for their thoughtful and significant recommendations. The material appears to be much clearer now in its both purposes and conclusions. Especially considering the characterization of the two periods and the debate on stability.

Thank you for your help in improving the manuscript.

Regarding this first point, I find that the objective of the paper is now clearer. There is no longer any ambiguity where the previous wording might have suggested that the objective was to see how limited episodes of MHW could have impacted the food web. It is now clear that the aim is to compare two periods, a reference period before the first, historical MHW episode and a prolonged period of thermal anomalies marked by at least 2 referenced MHWs (or Blobs). On this point, however, I think the authors could have been even clearer, for example at the end of page 4, by specifying that the recent period is a prolonged period of thermal anomalies marked by at least 2 referenced MHWs. A distracted reader might think that the whole period (8 years) is considered a single episode of MHW, but that's not how it's described in the literature. I feel it is

all the more necessary to insist, otherwise the name of the period (MHW period) could once again lead to confusion.

Thank you for this comment. We have added this language to the end of page 4 to read: “Here, we compare two end-to-end ecosystem food web models of the Northern California Current representing time periods immediately preceding (1999-2012) and following (2014-2022) **the onset of a prolonged period of thermal anomalies (Fig. 1a) marked by at least two well-described marine heatwaves**^{36–38} to make inferences about ecosystem-level changes that have occurred since the onset of these recent extreme warming events” per your suggestion.

Text earlier on the page also defines at least two MHWs during this period of anomalously warm conditions: “The **initial MHW** starting in 2013 included temperatures roughly 3 °C above normal (exceeding three standard deviations) and persisted in two prolonged pulses until 2015^{18,19}. **Re-occurring MHWs, including the so called blob 2.0 in 2019**, have since kept much of the North Pacific Ocean in a state of anomalously warm conditions over the past decade, indicating that these novel conditions are perhaps the new normal^{18–20}.”

We believe that the “MHW period” is clearly defined within these two paragraphs, suggesting that this is a period of anomalously warm conditions, marked by at least two well-described marine heatwaves, such that there will likely be no confusion. Thank you for helping to clarify this.

Another small but revealing detail, p5, following my prior comments, the authors notably reworded the section on TL, acknowledging in their response that average TL is not so directly related to chain length. However, there is still some ambiguity because the measure they used is average TL and the conclusion they draw is about the number of trophic levels. Both measures are still not identical. I admit that it isn't particularly important to the subject, but I believe it is always best to be exact in what you say.

Thank you for this comment. We disagree that we acknowledged that average TL is not related to chain length. Instead, we argued in the last round of revisions that these two things *are directly related*, yet not the same metric:

“...Borrelli & Ginzburg (2014) address food chain length, but, per their definition, this is inextricably linked to trophic levels “Food chain length is directly related to the number of trophic levels in a food web. A food chain with two steps has three trophic levels...” (Borrelli & Ginzburg, 2014). With that said, we agree that the maximum chain length and average trophic level is not necessarily the same thing. We’ve altered the language around this to better reflect the Borrelli and Ginzburg reference. We agree that TL estimation is contingent upon the aggregation of trophic groups, however, in our model comparison trophic groups are defined the same way such that average TL estimation differences should only be based on differences in biomass and diets between the two models.”

Mean, or average, trophic levels are one of many possible metrics used to understand networks. While Borrelli & Ginzburg focus on longest chain length, the other references cited within this section in the manuscript (refs 47, and 48, below), both use mean trophic levels as metrics to assess stability or efficiency:

Funes, M., Saravia, L. A., Cordone, G., Iribarne, O. O. & Galván, D. E. Network analysis suggests changes in food web stability produced by bottom trawl fishery in Patagonia. *Scientific Reports* 12, 1–10 (2022).

Olivier, P. et al. Exploring the temporal variability of a food web using long-term biomonitoring data. *Ecography* 42, 2107–2121 (2019).

And there are many others. E.g.,

Fu, C. et al. Making ecological indicators management ready: Assessing the specificity, sensitivity, and threshold response of ecological indicators. *Ecol. Ind.* 105, 16–28 (2019).

Su, L. et al. Decadal-scale variation in mean trophic level in Beibu Gulf based on bottom-trawl survey data. *Mar. Coast. Fish.* 13, 174–182 (2021).

“The marine trophic index measures the change in mean trophic level (MTL), and it can reflect the sustainability of fishing and the integrity of the marine ecosystem (Pauly et al. 1998, 2001; Jennings et al. 1999; Rochet and Trenkel 2003; Ji et al. 2010).”

We agree that it is important to be exact in what we say. Indeed, this is why we say average trophic level when we are discussing the metrics we measured and trophic levels, more generally, when discussing the theory, since there are multiple metrics used in the literature to assess stability and efficiency of networks.

Apart from these details, I think that the authors have answered my questions and comments in a relevant and well-argued manner. I hope the other reviewers will be as satisfied as I was, because I think this work deserves to be published.

Thank you for the positive feedback.